# Challenges in Management of Diabetic Patient on Dialysis

**Mohamed T. Eldehni** [1,2]**, Lisa E. Crowley** [1,2] **and Nicholas M. Selby** [1,2,*]

1 Centre for Kidney Research and Innovation, Academic Unit of Translational Medical Sciences, School of Medicine, University of Nottingham, Nottingham NG7 2RD, UK

2 Department of Renal Medicine, Royal Derby Hospital, Derby DE22 3DT, UK

* Correspondence: nicholas.selby@nottingham.ac.uk

**Abstract:** Diabetes mellitus is the leading cause of end-stage kidney disease in many countries. The management of diabetic patients who receive dialysis can be challenging. Diabetic dialysis patients have higher rates of cardiovascular events and mortality due to metabolic factors and accelerated vascular calcification. Diabetic haemodialysis patients have high rates of haemodynamic instability which leads to organ ischaemia and end organ damage; autonomic dysfunction seems to play an important role in haemodynamic instability and abnormal organ perfusion during haemodialysis. Poor glycaemic control contributes to fluid overload and worse cardiovascular outcome. Xerostomia and thirst are the main drivers for fluid overload in haemodialysis patients and in peritoneal dialysis a chronic state of hyperhydration that is related to absorption of glucose from the PD fluids, protein loss and malnutrition contributes to fluid overload. Glycaemic control is of great importance and adjustments to diabetic agents are required. In haemodialysis, a reduction in insulin dose is recommended to avoid hypoglycaemia whereas in peritoneal dialysis an increase in insulin dose is often required. Foot ulcers and infection are more common in diabetic dialysis patients compared to non-diabetic dialysis patients or diabetic patients with normal renal function and regular surveillance for early identification is important. Ultimately, a multi-disciplinary approach which includes diabetologist, nephrologist, dietitians, microbiologist, vascular surgeon, interventional radiologist is required to address the complicated aspects of diabetic patient care on dialysis.

**Keywords:** cardiovascular disease; myocardial stunning; intra-dialytic hypotension; advanced glycation end-products; arterio-venous fistula





## 1. Introduction

Diabetic kidney disease (DKD) is the leading cause of end-stage kidney disease (ESKD) in many countries [1]. The management of diabetic patients on dialysis can be challenging due to the high rates of complications in this patient group with factors such as peripheral and cardiovascular complications contributing to frequent hospitalisations and poor outcomes. In this review we summarise the key challenges in caring for diabetic patients on dialysis. This encompasses discussion of longer-term outcomes, including factors contributing to cardiovascular disease, as well as dialysis-related issues such as haemodynamic instability and vascular access in haemodialysis (HD), metabolic consequences of peritoneal dialysis (PD), plus aspects of glycaemic control and management of volume status.

## 2. Mortality, Cardiovascular Events

Historically diabetic dialysis patients have been found to have inferior survival compared to non-diabetic patients [2] although overall mortality rates in diabetics have declined in comparison to those at the beginning of the 21st century [3]. Rates of cardiovascular death are greatly elevated in all dialysis patients, and this is likely related to the complex interplay of classical and non-classical cardiovascular risk factors. Diabetic patients with chronic kidney disease due to diabetic nephropathy generally have co-existing multiple

micro and macrovascular complications but they also accumulate 'non-classical' risk factors associated with chronic kidney disease (CKD).

## 2.1. Metabolic Factors

One example of a 'non-classical' cardiovascular risk factor is the accumulation of tissue advanced glycation end products (AGE's). These are biomarkers of metabolic and oxidative stress which correlate, in the general diabetic population, with the presence of diabetic complications [4] and have been found to predict cardiovascular complications and mortality in HD patients [5]. Grossly elevated AGE levels have been found in both HD and PD patients with levels in the latter group associated with time since dialysis initiation and historical glucose exposure [6]. A recent meta-analysis of 11 studies found a significantly higher level of AGE accumulation in diabetics as compared with non-diabetics, particularly HD patients [7]. There are now a number of non-invasive methods available to measure skin AGE accumulation, although care needs to be taken with measurements in non-Caucasians and with measurements from the fistula arm [8]. From a clinical perspective, these measurements could be used to stratify patients at greater cardiovascular risk. AGE values measured by skin autofluorescence appear to increase over time in dialysis patients and these increasing values appear linked to smoking and nutritional factors [9,10]. Malnutrition may be a more important association than dietary intake of foods that cause predisposition to AGE formation. Therefore, theoretically, targeted dietary support could stabilise AGE levels [11]. There also continues to be great interest in developing anti-AGE therapeutics with attention focused on the inhibition of receptors for advanced glycation end products (RAGE) [12,13]. However, no specific therapies have been trialled as yet.

Accelerated vascular calcification is highly prevalent in haemodialysis patients [14,15], and diabetes per se is also associated with higher levels of vascular calcification [15–17]. Vascular calcification is associated with arterial stiffening and both are predictors of cardiovascular mortality [18]. Prevention of this phenomenon of accelerated calcification has proven very difficult. Plausible-sounding strategies such as avoidance of hyper-phosphataemia [15] and calcium-containing phosphate binders [19] have not demonstrated any effect. In recent years, attention has moved away from simply concentrating on the calcium–phosphate axis towards proteins involved in bone metabolism. There has been interest in vitamin K and its potential role as an inhibitor of vascular calcification with the hypothesis that CKD patients may have subclinical vitamin K deficiency [20]. This has potential implications for the use of Warfarin in end-stage renal disease patients. It is possible that this change in emphasis in the approach to vascular calcification research leads to more robust prevention strategies, although at present there no effective interventions for either preventing, slowing or reversing this process.

## 2.2. Dialysis Modality

Given the high cardiovascular mortality and the association between haemodialysis and dialysis-induced cardiac injury [6] it is reasonable to ask whether peritoneal dialysis confers any kind of survival advantage. Studies comparing survival rates in diabetics undergoing PD versus HD have been conducted to evaluate this, but these are observational in nature and have mostly illustrated complexity of this question. A retrospective analysis of Dutch registry data determined that in the initial period after dialysis initiation the hazard ratio for mortality was less in all patients on PD. This survival advantage was greater in younger non-diabetics and diminished in older diabetic patients. At 15 months after dialysis initiation this survival advantage had gone and in older patients PD was associated with a higher mortality risk (although it is likely that many confounding factors are at play that cannot be accounted for in an observational study). This early survival advantage was attenuated somewhat by diabetes and age, something which has been demonstrated in other retrospective studies [2,21,22]. While HD has notable haemodynamic consequences that may drive cardiovascular mortality, PD has a number of other potential mechanisms

driving cardiovascular risk. These may include excessive absorption of glucose, inadequate volume control and hypokalaemia [23].

Vonesh et al. added their own additional analysis of US Medicare data to the evidence from the Canadian, Danish and Dutch registries. They concluded that there were differences in case mix adjustment and sub-group analysis between the studies, but even when accounting for this there were a number of important similarities. PD was generally found to be associated with better survival (at least in the first 2 years) in younger diabetic patients [24]. However, a systematic review conducted for the European Renal Best Practice in Diabetes group concluded that "The available evidence derived from observational studies is inconsistent. Therefore, evidence-based arguments indicating that HD or PD as first treatment may improve patient-centred outcomes in diabetics with ESKD are lacking. In the absence of such evidence, modality selection should be governed by patient preference, after unbiased patient information" [25].

### 2.3. Primary and Secondary Cardiovascular Prevention

The evidence for pharmaceutical therapies to modify the enormous cardiovascular risk of a diabetic dialysis patients is also very limited. Notably, there has not been any benefit shown with the use of lipid lowering therapies for primary prevention in this group [26] although many diabetic dialysis patients continue to be prescribed statins for both primary and secondary prevention. There is uncertainty about how to manage blood pressure in the overall dialysis population as extremes of blood pressure are associated with unfavourable outcomes for all dialysis patients [27]. There also remains uncertainty regarding the optimal HbA1C target to aim for and how tightly to control blood sugars in dialysis patients. Multiple large cohort studies in HD patients suggest a U-shaped relationship between HbA1C and mortality with levels less than 6% and greater than 9% being associated with increased risk of mortality [28–30]. It remains possible that the lower HbA1C levels in these cohorts were markers of malnutrition and illness rather than adverse events causally related to tight diabetic control. While the evidence remains observational in nature, these studies in HD patients and others conducted in the PD population suggest that poor glycaemic control is also associated with worse outcomes [31]. Persistent hyperglycaemia may be associated with higher inter-dialytic weight gains and worsening macro- and microvascular complications and therefore striving to improve glycaemic control is an important aspect of care despite the lack of evidence that it leads to improved outcomes overall.

Cardiovascular mortality in all dialysis patients continues to be one of the most important clinical issues in nephrology. This is exacerbated further in diabetic dialysis patients. Current evidence does not point to a definitive survival advantage with either dialysis modality. The impact of different modalities in particular haemodialysis versus haemodiafiltration on cardiovascular outcomes and survival remains debatable, and the meta-analyses of published trials were in the end incapable of definitively answer whether there is a clear benefit from choosing one modality over the other [32]. Furthermore, studies that compares the effects of the two modalities in diabetic haemodialysis patients are lacking.

Beyond this there is considerable uncertainty, the available evidence suggests that maximising nutritional status and individualizing blood pressure (BP) and HbA1C targets to avoid extremes is best practice, but a survival benefit is difficult to demonstrate. The non-classical risk factors associated with overall and cardiovascular mortality continue to be potential targets for improved therapeutic options.

### 3. Vascular Access

In all HD patients the formation of timely and secure vascular access is a key component in patients' overall care. Central venous catheter (CVC) usage is associated with a higher mortality than in patients who start with an arteriovenous fistula (AVF). CVC use is also a driver of infection and hospitalization. Therefore, an arteriovenous fistula is, for the

majority of patients, the vascular access of choice. However, formation of an AVF can fail due to thrombosis and a failure to mature in all CKD patients.

In the 1980s there were reports of higher failure rates in diabetic patients undergoing a radiocephalic fistula formation compared with non-diabetics. Significantly lower patency rates were reported at one and five years leading to the historical suggestions that AVFs were not necessarily the first-choice vascular access in elderly diabetic patients [33]. However, there are other reports of better results in creating access for even elderly diabetics although the incidences of steal syndrome and thrombosis were still found to be higher [34], while other groups reported higher rates of success in diabetic patients when forming brachiocephalic as opposed to radiocephalic fistulas [35].

Cohort studies support the early evidence of higher rates of fistula failure in diabetics. In 317 AVFs created at the Mayo clinic between 2006 and 2008, diabetes was associated with reduced primary patency (Odds Ratio 1.54) [36]. A much larger registry-based retrospective study of haemodialysis patients in France also suggested the presence of diabetes may be associated with problems in the formation of timely vascular access. The study of patients in the Renal Epidemiology and Information Network (REIN) registry who started HD between 2005 and 2013 found that non-functional access at HD initiation was more common in diabetics than non-diabetics. Sub-group analysis by presence of co-morbidities suggested higher rates of functional access at dialysis initiation in healthier HD patients with fewer co-morbidities compared with diabetics and those with two or more cardiovascular co-morbidities. It also suggested that the rate of non-functional access in the sub-groups levelled out but with a bigger time lag between AVF formation and HD initiation. In other words, co-morbidity had a smaller effect on the rates of non-functional access if the fistula was made more than 6 months before initiating HD [37]. This points to rates of fistula maturation being slower in those patients with diabetes but still potentially successful, which is an important observation for pre-dialysis planning. Other studies looking at factors predicting AVF failure also suggest diabetes as an independent prognostic factor [38,39].

There is, however, conflict in the literature. Development of a risk equation to predict AVFs that fail to mature did not identify diabetes as a risk factor when formulating their model [40]. This is consistent with other studies [41–43] that failed to show a specific association between diabetes and a failure to mature. These conflicting findings can be partially explained by the lack of robustly designed prospective studies that comprehensively explore all aspects of fistula formation. Almost all of the above-quoted studies are retrospective in nature.

There remains much controversy about how to increase the rates of successful AVF formation and promote AVF survival. Okomuro et al. suggested that careful selection criteria using clinical examination and vascular mapping via ultrasound could lead to the successful creation of radiocephalic fistulas in diabetic patients and reported AVF survival rates no different to non-diabetics [44]. A retrospective study in the US suggested that diabetes was not a factor in delayed fistula maturation and that more attention should be paid to pre-operative arterial diameter (rather than the usual standard of a venous diameter > 2.0 mm). This would naturally lead to the placement of higher numbers of brachio-cephalic fistulas [45].

The best path to successful fistula formation in diabetics would seem to lie in doing the simple things well. Early referral, experienced surgical support with a multi-disciplinary team and careful pre-operative assessment currently represent the best approach along with adequate post-operative care and a proactive approach to AVF monitoring on the dialysis unit. There is no convincing evidence that we should be overly pessimistic about the likelihood of successful fistula formation in patients with diabetes although we should bear in mind that there may be issues with fistula maturation, and this may take longer. Development of robust predictive measures to help with selecting appropriate sites for fistula formation may be helpful but further work is needed to evaluate this.

## 4. Haemodynamic Instability during Dialysis

Haemodynamic instability is a specific complication of HD as opposed to PD. Patients with diabetes have higher rates of intradialytic hypotension (IDH) [46], and there are multiple predisposing factors that underlie this including autonomic dysfunction, higher inter-dialytic weight gains (IDWG) due to poor glycaemic control and vascular calcification [47,48]. Haemodynamic instability predisposes these patients to higher rates of organ ischaemia and damage. Myocardial hypoperfusion during haemodialysis leads to myocardial stunning which causes areas of regional wall motion abnormalities [49]. This exacerbates intradialytic hypotension and is associated with higher rates of heart failure and mortality [50]. Acute ischaemic brain white matter changes have also been demonstrated during haemodialysis [51]. This now well-documented process of dialysis-induced end-organ injury can be ameliorated in the general HD population by interventions that reduce IDH, of which the strongest evidence exists for the use of cooled dialysate [51–53]. One factor that may be particularly relevant to patients with diabetes is autonomic dysfunction, which seems to play and important role in the development if intradialytic hypotension. Cooling the dialysate has been shown to have beneficial effects on haemodynamic stability, brain white matter structure and cardiac function [51,52,54]. The evidence related to cooling the dialysate in diabetic haemodialysis patients remains very limited but suggests a similar beneficial effect on haemodynamic stability [55]. Using continuous blood pressure monitoring, analysis of frequencies of variation between peaks and troughs of mean arterial blood pressure has shown that higher frequencies correlate with increased organ damage and seems to improve with dialysate cooling [51,56,57]. More recently, it has been shown that groups of patients categorised by the ratio of high to low frequency variation in intra-dialytic blood pressure were characterised by different haemodynamic responses to HD. In those with greater lower frequency variation, blood pressure was more dependent on cardiac function, whilst in those with higher frequencies, blood pressure was more dependent on increasing peripheral resistance that suggests reduced cardiac reserve [58]. In future, novel approaches to continuous monitoring of blood pressure during dialysis (which would also allow a more pragmatic approach to assessing blood pressure variation) may offer opportunities for earlier detection, prediction and intervention to reduce the frequency and severity of IDH episodes [59].

## 5. Glycaemic Control and Drug Adjustments after Starting on Dialysis

In both PD and HD, patients often require adjustments of insulin at the start of dialysis. These adjustments differ between the two modalities. Hypoglycaemia is common in diabetic patients on haemodialysis; one study found this to be as high as 46.6% [60]. Glucose levels drop during haemodialysis; a study using continuous glucose monitoring found that 16% of patients had asymptomatic hypoglycaemia during dialysis that was not detected by finger prick test [61]. Hypoglycaemia in haemodialysis patients is likely due to several different factors. As insulin is metabolised in the kidney, the progression of CKD to ESKD requires a reduction in insulin dose, especially in haemodialysis patients. Furthermore, insulin is a low-molecular-weight peptide and in theory can be removed or adsorbed by a high-flux membrane. In a study that compared insulin removal between three high flux dialysis membranes polysulfone, cellulose triacetate, and polyester polymer alloy found that all three types of membranes reduce insulin levels, but the highest reduction was polysulfone [62]. At the same time, dialysate has a lower glucose concentration than the blood and this leads to removal of glucose from the blood during HD. Moreover, adjustments to diabetic oral agents are required when starting on dialysis, as summarised in Table 1.

**Table 1.** Recommended dose adjustments in diabetes medications at start of dialysis.

|  | **Haemodialysis** | **Peritoneal Dialysis** |
|---|---|---|
| Insulin | Reduce by up to 25% | Increase by up to 30% |
| Metformin | Not recommended | Not recommended |
| Sulfonylurea |  |  |
| Linagliptin | 5 mg/od | 5 mg/od |
| Sitagliptin | 25 mg/od | 25 mg/od |
| Vildagliptin | 50 mg/od | 50 mg/od |
| Alogliptin | 6.25 mg/od | 6.25 mg/od |
| SGLT2 Inhibitors | Not recommended | Not recommended |
| GLP1 Receptor Agonists | Not recommended | Not recommended |

Peritoneal dialysis is different in that there is an excess of glucose that is absorbed during the dialysis process and adjustment of the insulin dose is required to maintain glycaemic control. Glucose absorption has metabolic effects such as insulin resistance, dyslipidaemia, obesity and coronary artery calcification [23]. A systematic review of the literature looking at insulin adjustment recommendations for HD and PD found that the most common recommendation for HD patients is to reduce the basal insulin dose by up to 25% on HD days to prevent hypoglycaemia [63]. Little information and consensus were found when it came to PD but an increase of up to 30% may be required to mitigate the effects of dextrose absorption for the peritoneal dialysis fluids. A systematic review found that neutral pH and low-glucose-degradation peritoneal dialysis solutions improved the preservation of residual renal function compared to glucose PD solution [64]. The use of Icodextrin solutions reduced episodes of fluid overload and improved ultrafiltration without compromising residual renal function [64].

## 6. Fluid Management in Diabetic Dialysis Patients

Fluid overload in both haemodialysis and peritoneal dialysis has been found to be linked with increased adverse cardiovascular outcomes and increased mortality [65,66].

In haemodialysis patients the recommended intradialytic weight gain is less than 4–4.5% of dry weight [67]. An intradialytic weight gain of more than 4% has been identified as an independent predictor of all-cause mortality [68]. However, it is estimated that around 30–60% of haemodialysis patients do not manage to adhere to an optimal fluid restriction regimen [69]. Interdialytic weight gain is caused by excessive fluid intake, driven by increased thirst and xerostomia. Xerostomia is very common in haemodialysis patients with an estimated prevalence of 28.2–66.7% [70,71]. Salivary flow is significantly reduced in haemodialysis patients as a consequence of atrophy and fibrosis of the salivary glands [70]. Thirst in haemodialysis patients is also linked to both sodium and glucose levels, which is particularly relevant to diabetic patients on haemodialysis. Thirst and xerostomia score are significantly higher in diabetic haemodialysis patients compared to those without diabetes [72]. There is currently no definite treatment for xerostomia in haemodialysis patients. Several treatments and techniques have been trialled with variable results. Chewing gum was not associated with increased salivary production in haemodialysis patients [73]. Electrical stimulation of the salivary glands could restore normal salivary flow and improve patients symptoms but had no effect on intradialytic weight gains [74]. Combining auricular acupuncture and fluids restriction regimen in one randomised controlled trial led to improved salivary flow, better fluid status control and reduced intradialytic weight gains compared to the control group [75].

Hyposalivation and higher intradialytic weight gains were also found to be higher in diabetic patients, and higher levels of HbA1c were associated with lower levels of plasma sodium [72]. Higher levels of HbA1c were also found to be associated with higher blood

pressure and larger intradialytic weight gains compared to patients with levels of HbA1c of less than 6% [76]. The same study found that 70% of patients with a HbA1c >8% required antihypertensive medication [76]. Hence, salt restriction and good glycaemic control is of great importance in reducing intradialytic weight gains and consequently improving cardiovascular outcomes, reducing IDH and ultimately impacting survival in diabetic haemodialysis patients.

Hyperhydration and fluid overload is also common in PD patients, affecting between 53.4% to 72.1% of patients [77], and is more common in patients on PD as compared to HD [78]. The aetiology of hyperhydration in peritoneal dialysis is multifactorial. Similarly to haemodialysis, diabetic PD patients have a higher incidence of fluid overload than non-diabetic patients [79]. Peritoneal protein loss, poor nutritional status and loss of muscle mass are also factors implicated with hyperhydration in patients on PD and is associated with higher mortality [65]. The use of bioimpedance to assess overhydration in PD to guide management has been proposed by several studies [80,81], with bioimpedance-guided clinical decisions and peritoneal dialysis prescription found in one study to improve fluid overload but had no effect on one year survival or cardiovascular events [82]. One randomised study tested the use of Tolvaptan in diabetic peritoneal dialysis patients with difficult fluid control and demonstrated improvement in urine output, fluid status and preservation of residual renal function in the Tolvaptan group [83]. This is not yet an established treatment as it requires adequate residual renal function and stronger evidence from larger scale trials.

## 7. Nutritional Considerations

Malnutritional in dialysis patients is an important clinical problem. Restrictive renal diets (including low-protein diets), CKD-induced inflammation and catabolism, and loss of nutrients via dialysis are all contributing factors [84,85]. Malnutrition scores and low serum albumin levels are independent predictors of mortality [86] and a lower quality of life [87].

Diabetic dialysis patients tend towards having a higher BMI, however there are multiple studies suggesting that they are concurrently at increased risk of malnutrition. A cohort study from France included 170 diabetics and found that in comparison to those in the non-diabetic cohort there was a lower serum albumin and lean body mass in the diabetic cohort. The study did suggest that the decreased survival it found in diabetics was not related to malnutrition as only age was an independent mortality predictor [88]. A representative survey of HD patients in Israel found that despite a higher BMI in the HD population, there was a greater risk of malnutrition compared with non-diabetics. When modelled alongside other factors, diabetes more than doubled the odds of malnutrition [89].

There has long been a suggestion of an obesity paradox in HD patients where a high BMI is associated with a better survival [90]. The evidence for this has been consistent amongst several cohorts, although this paradox has not been demonstrated in PD patients. Given that BMI is not necessarily an accurate indicator of nutritional status and that higher BMI in diabetic patients appears to be associated with an increased risk of malnutrition, healthcare professionals should ensure that they are not misled by an elevated BMI and that full assessments of malnutrition risk are undertaken. Integrating more sophisticated methods of body composition into our assessments could also be of benefit [91].

## 8. Quality of Life Measures

As survival of end-stage renal failure patients on dialysis has improved, health-related quality of life (HRQoL) has increased in importance. HRQoL may be a predictor of mortality in ESKD patients [92]. The data on HRQoL in diabetic dialysis patients versus those without diabetes is not comprehensive. A Japanese study of 527 HD patients found that the baseline physical component score of the SF-36 questionnaire was an independent risk factor for mortality although the mental health component was not [93]. Conversely, Lopex Revuelta et al. enrolled 208 diabetic patients in a similar study and found that both the physical and

mental components of the SF-36 predicted mortality in diabetic dialysis patients [94]. A Norwegian group hypothesized that HRQoL in diabetic dialysis patients was at least as poor as those with another severe complication such as foot ulcers. Their cross-sectional study of prevalent dialysis patients (again using the SF-36) with and without diabetes found lower perceived physical health than non-diabetics on dialysis or patients with diabetes but not ESKD. Mental health aspects were independent predictors of mortality in dialysis patients with diabetes [95]. The above findings underline the challenges as the diabetic dialysis population grows in size. Improving the outcomes and quality of life of those patients with multiple, severe co-morbidities requires a holistic approach to physical and mental well-being.

## 9. MDT Approach to Diabetic Dialysis Patients

Management of diabetic dialysis patients on HD and PD requires a holistic multi-disciplinary approach. Diabetic patients on dialysis have higher incidence of diabetic foot ulcers, lower-limb amputations and infections compared to non-diabetic dialysis patients and diabetic patients with normal renal function [96]. Adding together the complex nutritional, cardiovascular, vascular access and dialytic aspects, an approach that includes a diabetologist, nephrologist, renal dialysis nurses, dietitian, microbiologist, vascular surgeon and interventional radiologist is required to cover all of the complicated aspects of diabetic patients care on dialysis.

## 10. Conclusions

Diabetes adds complexity to the management of patients on dialysis. Challenges are related to increased mortality and cardiovascular morbidity, higher rates of infections, fluid overload, dialysis-related complications, psychosocial and quality and life. A holistic approach to all these challenges through a multidisciplinary team approach is most likely be effective in the management of these patients.

**Author Contributions:** Conceptualization, M.T.E., L.E.C. and N.M.S.; resources, M.T.E. and L.E.C.; writing original draft preparation M.T.E. and L.E.C.; writing—review and editing M.T.E., L.E.C. and N.M.S. All authors have read and agreed to the published version of the manuscript.

**Funding:** This research received no external funding.

**Institutional Review Board Statement:** Not applicable.

**Informed Consent Statement:** Not applicable.

**Data Availability Statement:** Not applicable.

**Conflicts of Interest:** The authors declare no conflict of interest.

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
