# Peer review of "Challenges in Management of Diabetic Patient on Dialysis"

_kidneydial, doi:10.3390/kidneydial2040050_

Round 1

Reviewer 1 Report

Eldehni MT and coworkers made a review on the challenges in the management of dialysis diabetic patients with some proposals for improving outcomes in this vulnerable population. In addition, they analyzed relative benefits and risks associated with the use of hemodialysis or peritoneal dialysis. It is a well and clearly written narrative comprehensive review addressing most of challenges and proposing practical approaches (metabolic and fluid control, dialysis strategy) to mitigate additional risks associated with diabetes in dialysis dependent patients.

My only suggestion would be to complete this review by adding further documented comments on missing information.

1.             Hemodialysis modality including treatment schedule and various technical options will benefit from more details:

a.             Treatment time and frequency. Longer treatment time (daily or nocturnal) is beneficial to reduce cardiovascular stress (reduced ultrafiltration rate, better fluid volume and hemodynamic tolerance).

b.             High volume hemodiafiltration is associated with better control of uremic toxins including middle molecular compounds (B2), including oxidative and carbonyl stress, AGEs, inflammation markers … higher hemodynamic stability…. and associated with a reduction of cardiac mortality (HDF pooling project, Peters, NDT; Davenport, KI)

c.              Hypothermic dialysis and ultrafiltration-controlled (volemic preservation) dialysis are also associated with better short term and midterm outcomes.

d.             Dialysate electrolytic prescription should be discussed. In particular sodium, calcium, magnesium, potassium and bicarbonate and/or acidifier (citric ac/lactic ac…) since they are associated cardiac risk (arrhythmia), calcification and cardiovascular risk

e.             Preservation of residual kidney function through treatment schedule adjustment (incremental dialysis) or hemodynamic stability is also very important.

f.         Fluid volume assessment, monitoring (multifrequency bioimpedance or lung US), including blood pressure control is of tremendous importance in this vulnerable population.

2. Vascular access

a. Vascular access creation: additional details are needed to assess arm and forearm vascular network to prevent distal ischemia or cardiac consequence with proximal AVF creation with usually calcified arteries.

b. Vascular access monitoring would also be interesting to comment since they are more exposed to dysfunction.

3. Peritoneal dialysis deserves also further comments:

a.             Automated PD versus CAPD: is there any benefit from one method to the other?

b.             Dialysis dose delivery including residual kidney function: what weekly dose?

c.              Preservation of peritoneal membrane functionalities: what role for biocompatible, bicarbonate buffered, low GDP solutions… 

d.             Fluid volume control: Patient monitoring, Peritoneal ultrafiltration, use of icodextrin solutions, low sodium solutions

4. Specific monitoring and frequency of diabetic dialysis patients should be discussed little bit more: ophthalmologic, cardiovascular in particular to detect silent ischemia, neurologic…

Author Response

We have added some discussion about HH, HDF and the effects of cooling the dialysate. (Page 6 and Page 8)

We have addressed the issues related to discussing other Non-glucose peritoneal dialysis fluids. (Page 9)

Reviewer 2 Report

    This manuscript is well-prepared and well-written. The contents are mostly fine and acceptable. Please consider to describe some aspects about the condition (chances、management、outcome...) of transplantation for these patients, even though it is not exactly to the topic of this article.

Author Response

We have added some discussion to hospitalisation.

Reviewer 3 Report

This review focuses the issues/ challenges from the physician’s perspective

(dialysis consideration needs to be inclusive of both community and home dialysis and each comes with special challenges for DM population)

Other challenges to be address relating to DM patient on dialysis for physicians should also include :

1 .visual impairment on DM population on dialysis

2. assistance for dialysis (amputee / visual impairment): transportation

3. need for caregiver in home dialysis

4. DM dermopathy: relevant in exit site infection for HD and PD catheters

5. Polypharmacy and medication burden in this group 

6. on holistic management of patient, author need to also focus on the mental health or patient related outcome in DM patients on Dialysis: any evidence of that they have poorer QoL ?

To conclude;

The reader would be better served if the article covers some aspects of

Challenges these patients present to the overall healthcare system, perhaps focusing on hospitalization, QALYs and comments on how some of these challenges can be mitigated.

The author should have a brief mention / consideration of extra challenges of this group of patients in all other healthcare settings esp if healthcare financing is not universal e.g. LMIC etc, from physician perspective are there any ethical challenges

Review article need to be encompassing

Mortality cardiovascular events

Line 39

Can the author quote mortality risk of DM patients on HD /PD or dialysis overall to give reader an impression of severity

Dialysis modality

Line 82

In this paragraph of modality comparison, can the author consider geographical differences in particular in Asia , suggest to be more globally inclusive

Line 110:

This paragraph has summarized the challenges of lack of clinical target to treat, would suggest to make some references to KDIGO /KDOQI guidelines where relevant and to give some “optimism” to this section

Vascular access

Line 138 :

I note that the vascular access section is done with some solutions of early access creation and increase monitoring. This section is proportionately larger than other sections and author to select more pertinent and relevant study to the review

Line 140:

can author provide reference for this statement (Jeff Perl has a paper on this)

Hemodynamic instability

Line 195:

This paragraph describes IDH in HD population and that these symptoms and signs are exaggerated due to concurrent co-morbidities, offering some solution and explanation. This paragraph also offers some insight into future mitigation methods e.g. continuous BP monitoring during dialysis. I would suggest also with technological innovation and data science that author could look into AI for prediction as well.

Line 196:

The first sentence with regards to “specific complication of HD” , gives the impression that it does not occur in PD ?  suggest to rephrase. Excess UF during CAPD can occur   though the quantum may be less.

Glycaemic control:

Line 224:

knowing the challenges, is there any way one could reduce these events of hypoglycaemia?  How do dialysis centre manage this? increase monitoring? eat before / during dialysis, increased glucose in dialysate or reduce insulin etc. Some mitigating solutions would be helpful for readers.

Line 232:

suggest nomenclature change from ESKD instead of ESRD

Line 250: 

Table on medication dosing adjustment: please insert reference below the table 

Line 251:

Fluid management:

Line 259:

Thirst in HD maybe related to excessive and rapid ultrafiltration during HD, poorly control DM leading to hyperosmolarity. The latter is more pertinent in PD as this will reduce the osmotic gradient for UF in PD.

Line 292:

Ref 79 is a pilot study, suggest not to use the word “randomized “. Study arm are patients recruited to study after regulatory drug approval, control arm from before approval ..drug

Nutritional considerations:

Paragraph focused on protein energy malnutrition. DM patient will have an added level of restriction in addition to dialysis diet, compliance will be a challenge, perhaps more so for DM population , worth a mention and be supported by reference if any

MDT:

Line 326:

Consideration to add podiatrist, medical social worker, pharmacists as part of the team

Author Response

We found no significant literature related to diabetes dermatopathy in relevance to HD and PD catheters.

We have added a discussion related to quality of life. (Page 12)

We have added some discussion to hospitalisation.

Reviewer 4 Report

The paper is interesting and it seems to be updated. In some points English scientific language can be improved and in other points there are tipewriting mistakes (see Abstract, lines 14, 20). Table 1 is uncorrectly defined as "...oral diabetes medication..." (please delete"oral") and it is not clear what one should do with sulfonylureas. Please also delete the "DPP4 Inhibitors" line

Author Response

Table 1 amended as requested

Round 2

Reviewer 3 Report

paper provided a good summary of problems /challenges treating DM patient with CKD.

there are some minor spelling and grammar errors, 

paper is length, would be good to have a summary table but not essential

Author Response

Please find attached the revised manuscript with all comments addressed.
